# Effects of Caffeine Intake on Self-Administered Sleeping Quality and Wearable Monitoring of Sleep in a Cohort of Young Healthy Adults

**DOI:** 10.3390/nu17091503

**Published:** 2025-04-29

**Authors:** Jenny Schlichtiger, Stefan Brunner, Anna Strüven, John Michael Hoppe, Christopher Stremmel

**Affiliations:** 1Department of Medicine I, LMU University Hospital, LMU Munich, 81377 Munich, Germany; jenny.schlichtiger@med.uni-muenchen.de (J.S.); strefan.brunner@med.uni-muenchen.de (S.B.); anna.strueven@med.uni-muenchen.de (A.S.); 2DZHK (German Centre for Cardiovascular Research), Partner Site Munich Heart Alliance, LMU University Hospital, LMU Munich, 81377 Munich, Germany; 3Department of Medicine IV, LMU University Hospital, LMU Munich, 80336 Munich, Germany; john.hoppe@med.uni-muenchen.de

**Keywords:** caffeine, energy drinks, wearable data, sleep impairment

## Abstract

**Background/Objective:** Chronic sleep deprivation can lead to adverse health effects and therefore represents a public health burden While caffeine is a widely used stimulant, the relationship between caffeine consumption and sleep remains uncertain. Moreover, many studies might be subject to methodological bias, as invasive sleep measurements may themselves serve as confounders. The aim of the SleepSmart study was to assess the impact of caffeine consumption through coffee and energy drinks, utilizing both smartwatch data and questionnaire responses. **Methods:** The SleepSmart study is an observational cohort study conducted at LMU Hospital Munich, Germany, from July 2024 to January 2025. The study investigated two intervention groups: (1) coffee and (2) energy drink consumption. Each intervention lasted 1 week, with days 1 to 3 serving as a reference period (water consumption to adjust for increased fluid intake) and days 4 to 6 serving as the intervention period (consumption of an equivalent of 240 mg of caffeine per day, 3 h before bedtime). Data was collected through self-administered questionnaires and wearable devices. The primary endpoint was sleep duration. Objective measures of sleep (duration of light/deep sleep, duration of wake phases, heart rate) and self-assed quality of sleep (Pittsburgh Sleep Quality Index) served as secondary endpoints. **Results:** For the primary endpoint, we did not find a significant difference in average sleep duration (baseline [hours] 6.7, coffee 6.9, energy drink 6.7, *p*-value 0.183). Similarly, secondary endpoints related to sleep measures showed no significant changes in the duration of light/deep sleep (coffee [min]: 242.2, reference coffee [ref_C_] 255.7 I energy drink: 237.5, reference energy drink [ref_E_] 261.3), time awake (coffee 22.7, ref_C_ 23.4 I energy drink 21.3, ref_E_ 18.9), time to fall asleep/wake up (coffee 4.2, ref_C_ 4.0 I energy drink 4.4, ref_E_ 2.7), or average heart rate (coffee [bpm] 62.2, ref_C_ 62.1 I energy drink 62.6, ref_E_ 62.3)—neither between the two intervention groups nor compared to the reference period. However, self-assessed sleep quality revealed a decrease in perceived sleep quality, with reports of poor sleep increasing by 8% after coffee consumption (rather bad/very bad sleeping quality: 29.7%) and 20% after energy drink consumption (46.6%), compared to baseline data (21.6%). **Conclusions:** The SleepSmart study introduces wearable sleep tracking as an innovative, low-interference method for objectively recording sleep data. While wearable data did not indicate significant sleep deterioration in the group of young, healthy adults, caffeine appeared to negatively impact the subjective perception of sleep in the study cohort.

## 1. Introduction

Sleep is crucial for long-term health, as chronic sleep deprivation is linked to numerous adverse health effects, including cardiovascular risk factors, depression, and a general decline in quality of life [1,2]. In addition to these long-term consequences, poor sleep patterns also have short-term effects, negatively impacting performance, concentration, and overall well-being [3,4]. Therefore, sleep is a multi-dimensional health factor that represents a significant public health burden, highlighting the need for a better understanding of its complexity in order to identify modifiable variables. Established risk factors for poor sleep include late-night eating, heavy meals, smoking, alcohol consumption, and a lack of exercise [5,6]. However, the impact of caffeine on sleep is still a topic of debate. While it is generally accepted that caffeine intake can lead to difficulty falling asleep and reduced sleep duration, the scientific evidence remains inconclusive [7]. Although several studies suggest that caffeine may impair sleep by reducing its duration and increasing sleep onset latency, it remains unclear whether these effects significantly impact individual health [8]. Moreover, many studies suffer from methodological issues, such as relying on self-reported sleep diaries, which may be influenced by individual perception, or using laboratory settings and polysomnography devices, both of which could potentially alter sleep patterns and introduce confounding factors. Additionally, most studies focus primarily on coffee intake, while other sources of caffeine, such as energy drinks and high-caffeine pre-workout supplements, which are particularly popular among young adults, are often underrepresented in current research [9].

Therefore, the aim of the SleepSmart study was to assess the impact of caffeine consumption from both coffee and energy drinks on sleep patterns using both smartwatch data and self-assessments via questionnaires. In line with the most recent and large-scale metanalyses on the effects of caffeine intake on sleeping patterns, which detected a reduction in sleep duration of 45 min by Gardiner et al., the sleep duration was prespecified as the primary endpoint within the SleepSleep Smart study.

## 2. Materials and Methods

### 2.1. Study Setting

The SleepSmart study is an observational cohort study conducted at LMU Hospital, Munich, Germany from July 2024 to January 2025, aiming to investigate the impact of caffeine intake on sleep in two intervention groups:Intervention Group 1: Caffeine intake via coffee consumptionIntervention Group 2: Caffeine intake via energy drink consumption

In both groups, participants had to consume an equivalent of 240 mg of caffeine three hours before going to sleep. Each intervention lasted for 7 days (see Figure 1). For both groups, days 1 to 3 served as a reference period, during which participants consumed water equivalent to their intake of coffee or energy drinks to adjust for changes in fluid intake. From days 4 to 6, participants consumed a dose of coffee or energy drink providing 240 mg of caffeine, taken three hours before bedtime [10,11]. To ensure that the required consumption values were met, participants were provided with a nutrition diary with a variety of coffee beverages or energy drinks, each with a specified caffeine dose, and were asked to mark the amount consumed and the time of consumption. If the daily consumption values were not met, participants were excluded from the analyses (no drop-outs due to non-achievement occurred and no one exceeded the daily consumption dose). Participants did not have to adjust their usual caffeine intake (e.g., in the morning), which is represented as the baseline measure, and is considered a constant variable during intervention. Study subjects were allowed to self-assign for an intervention group to increase coherence and compliance. Participation in only one intervention group was permitted and had to be predefined before entering the study phase. If the intervention was terminated, drop-out was registered. On day 7, participants had the opportunity to charge the wearable device.

Data collection included:A self-administered questionnaire at baseline (Day 1) and on the last day of the intervention (Day 7).Data from the wearable device.

### 2.2. Study Population

The inclusion criteria for participation were:Age > 18 yearsNo pre-existing medical conditionsNo contradiction between daily caffeine consumptionSinged consent for study participation

40 participants were included in the study, of which 93% (*n* = 37) completed Intervention 1, and 78% (*n* = 31) completed Intervention 2. No drop-outs or loss to follow-up were registered. Exclusions occurred due to incomplete data.

### 2.3. Outcome Measures and Endpoints

Sleep was measured using two methods:Self-assessed questionnaires containing validated questions on:Smartwatch tracking (Withings ScanWatch)

The Withings ScanWatch detects sleep by wrist movement and heart rate (for detailed information see withings.com). The accuracy of heart rate tracking via ScanWatch is validated and especially accurate at rest. [12,13] Upon inclusion, participants were introduced to the correct use and positioning of the watch by a study investigator. The watches were fully charged at the start of the intervention and have a battery life of up to 20 days. This allowed participants to collect tracking data throughout the entire observation period. (For a detailed list of obtained variables/data see Appendix A: Outcomevariables)

To be included in the analysis, participants had to provide complete data from the baseline questionnaire, follow-up questionnaire, and smartwatch tracking.

#### Primary and Secondary Endpoints

The primary endpoint was defined as the duration of sleep. Secondary endpoints included objective sleep parameters (duration of light/deep sleep, time to get up, heart rate) and subjective sleep quality (PSQI).

### 2.4. Statistical Analyses

Statistical analyses were performed using SPSS version 29 (IBM Statistics, Armonk, NY, USA).

For the description of metric data, measures of central tendency and dispersion were used. Due to the relatively small sample size, comparisons of means were conducted using non-parametric tests. Categorical data were described using absolute values and proportions, and for bivariate analyses, the Chi-square test (χ^2^) was applied. Due to the exploratory design of the study, no prior sample size calculation was performed.

The SleepSmart study was approved by the ethics committee of LMU Munich, Germany (approval number: #23-0972). The raw data supporting the conclusions of this article will be made available by the authors on request.

## 3. Results

### 3.1. Demographics and Baseline Data

Overall, 60% of the participants were female, with a mean age of 31 years. At baseline, 92% (*n* = 34) of the study cohort reported regular coffee consumption over the past four weeks, with an average daily intake of 285 mL (approximately 230 mg of caffeine per day). Additionally, 23% of participants reported regular energy drink consumption, with an average intake of 1.5 L per week (approximately 480 mg of caffeine per week). Prior to the intervention, 68% of participants reported that they experienced sleep problems regularly.

### 3.2. Primary Endpoint: Duration of Sleep (See Table 1)

For the primary endpoint, no significant difference in average sleep duration was observed (Baseline *[hours]* 6.7, Coffee 6.9, Energy drink 6.7, *p*-value 0.183).

**Table 1 nutrients-17-01503-t001:** Primary outcome Duration of sleep.

	Baseline	Coffee	Energy Drink	Baseline vs. Coffee vs. Energy Drink *p*-Value
Duration of sleep				
mean ± SD [hours]	6.9 ± 1.0	6.9 ± 1.3	6.7 ± 1.1	0.183

SD: standard deviation; metric data: Wilcoxen Test Statistic; Bonferroni correction for multiple testing was performed.

### 3.3. Secondary Endpoints

Sleep measures from wearable data (see Table 2A)

Interventions vs. Reference Period (Days 1 to 3): In both groups, the proportion of deep and light sleep phases was reduced during the intervention compared to the reference period (Coffee [min]: 242.2, Ref_C_ 255.7 I Energy Drink: 237.5, Ref_E_ 261.3). Likewise, the mean time awake during the night decreased through caffeine consumption (Coffee 22.7, Ref_C_ 23.4 I Energy drink 21.3, Ref_E_ 18.9). However, only the intake of energy drinks resulted in a prolonged time to wake up (Coffee 4.2, Ref_C_ 4.0 I Energy drink 4.4, Ref_E_ 2.7); while no difference in time to fall asleep compared to the reference period (Coffee 2.8, Ref_C_ 2.9 I Energy drink 2.5, Ref_E_ 2.4) was detected.

Coffee Intake vs. Energy Drink: A comparison between the two intervention groups revealed a higher proportion of deep sleep and a correspondingly lower proportion of light sleep in the Energy drink group (light sleep [min]: Energy Drink 237.5, Coffee 242.3 I Deep sleep: Energy Drink 192.3, Coffee 189.3), as well as a slightly lower time awake during the night (Energy drink 21.3, Coffee 22.7).

No differences were observed in time to fall asleep, time to wake up, or average heart rate between the two intervention groups and their corresponding reference periods.

2.Self-Assessment Data (see Table 2B,C)

Patients self-reported a prolonged time to fall asleep during both interventions (baseline [min] 23.5, Coffee 27.4, Energy Drink 27.3, *p*-value 0.183). Compared to the baseline, the proportion of participants reporting a deterioration in sleep quality (operationalized as “bad” or “very bad” sleep quality) increased by 8% following daily coffee consumption, and by 20% following daily energy drink consumption (see Table 3, Figure 2). Stratified analyses of the PSQI-questionnaire items showed that the proportion of participants indicating to “often” or “always” experience problems falling asleep was twice as high in the coffee group and three times higher in the energy drink group compared to baseline (Baseline: 14%; Coffee: 27%; Energy drink: 32%). Similar effects were observed for nighttime awakenings (Baseline: 5%; Coffee: 16%; Energy drink: 19%). However, despite a subjective decrease in recovery, the assessment did not show an increase in daytime sleepiness, sleep medication use, or difficulty managing daily routines.

3.PSQI Assessment (see Table 2C)

The Pittsburgh Sleep Quality Index (PSQI) showed an average of 7 in both intervention groups, indicating impaired sleep quality.

## 4. Discussion

In summary, the SleepSmart study demonstrated a reduction in both deep and light sleep phases. However, no significant differences were observed in the mean duration of sleep, time to fall asleep, or time to wake up when analyzing the sleep-tracking data collected via wearable devices. In contrast, the results from the self-administered questionnaire indicated that, during both interventions, participants experienced a decline in sleep quality, resulting in an increased proportion of participants reporting difficulties falling asleep, more frequent awakenings during the night, and a feeling of not being rested during the day. Overall, when assessing self-reported sleep quality, the PSQI scores for both intervention groups showed a moderate impairment of sleep, with no significant differences between the two interventions.

A recent meta-analysis by Gariner et al., which summarized the effects of caffeine intake on sleep patterns across 24 studies, quantified the impairment in sleep as a mean reduction in sleep duration by 45 min, an increase in sleep onset by 9 min, and an increase in wake after sleep onset by 12 min [8]. In line with these findings, the SleepSmart study observed a reduction in the average time of deep sleep in both intervention groups, and following the intake of energy drinks, a corresponding increase in nightly awakenings compared to the reference period. However, in contrast to Gariner et al., we did not observe a reduction in total sleep duration. These differences may be attributed to variations in the study cohorts, as sleep patterns and sensitivity to caffeine are strongly influenced by age [14,15]. Comparing our results to a study by Weibel et al., who investigated the impact of caffeine intake in young adult men (mean age 24 years) in a double-blind, randomized, crossover study involving caffeine, withdrawal, and placebo periods, the authors similarly found no significant differences in total sleep time, sleep latency, or sleep architecture across the three periods [16]. Another investigation in line with our results by Skala et al. on the effects of caffeine on sleep and physical activity in a cohort of older adults, also using accelerometer measures for data collection, found no association between caffeine intake and sleep efficiency. The authors hypothesize that no effect could be detected, as no time of intake was pre-specified in the study, and most participants confined caffeine consumption to early daytime [7]. Our results provide evidence that, at least in a cohort of young adults, even late consumption does not significantly affect objective measures of sleep. Additionally, the comparable heart rate profiles between the intervention and control groups may support the notion that caffeine intake did not have a substantial impact on the vegetative system, at least not as a mediating factor. Several studies have shown that, particularly in habitual coffee drinkers, caffeine does not cause a significant increase in heart rate [17,18].

Although we did not observe a significant impairment in sleep based on wearable data, participants reported a notable decline in sleep quality via questionnaire responses. The consumption of energy drinks appeared to have a more negative impact on subjective sleep quality compared to coffee. These findings are supported by a study by Claydon and colleagues, who investigated the effects of daily caffeine consumption, diet, and exercise as risk factors for poor sleep quality among university students. The study showed that caffeine intake from sodas was linked to a significant decrease in sleep quality compared to coffee or tea. [19] The adverse effect of energy drinks might be mediated by additional ingredients like sugar or supplements (e.g., taurine) that have the potential to amplify the effects of caffeine [12]. Furthermore, energy drinks were shown to increase the potential for anxiety and depression [20], which could be explanatory for the subjective decrease in subjective sleeping quality.

Within the SleepSmart cohort, we detected an inconsistency between the subjective perception of caffeine’s impact on sleep and the objective measure. Whilst the thePSQI score of 7, which indicates a poor sleep quality, was comparable between both intervention groups stratified analyses of the individual score items indicated that the perception of poor sleep quality was primarily due to difficulties falling asleep, increased wakefulness during the night, and poor recovery. However, we did not observe a relevant difference in the corresponding objective measures collected via a wearable. This contradiction could be attributed to perception bias or potential covariates—such as stress, diet, or mental well-being—that were not assessed within the study. This hypothesis is supported by the results of a study by Kerpershoek et al., who examined whether evening caffeine consumption moderates the relationship between caffeine intake and subjective sleep quality in college students. The investigators found that high caffeine intake was only associated with poor sleep quality in participants who did not consume caffeine in the evening. Furthermore, for those who reported poor sleep quality, caffeine intake was not associated with total sleep time or sleep-onset latency [21]. In line with our findings, these results may support the hypothesis that subjective perceptions of sleep quality may require a more complex model of potential risk factors, which cannot be captured solely by objective sleep measures.

The SleepSmart study aims to introduce an innovative method for collecting sleep data. Wearable data is of great clinical and scientific interest since it has the potential to serve as an early detection device of diseases or potential risk factors and facilitates high-quality data collection.

However, due to the relatively small sample size and the rather short intervention period as well as a lack of a control group data analyses are exploratory in nature and can only serve to generate hypotheses for future research. Studies in a randomized setting, including a control group are needed to determine if the described differences may be caused by a potential association of late caffeine intake and subjective vs. objective sleep quality. Furthermore, potential confounder variables like psychosocial and general nutritional behavior must be included to take into account the multidimensional entity of sleep.

## 5. Conclusions

The SleepSmart study introduces wearable sleep tracking as an innovative, low-interference method for objectively recording sleep data. Although caffeine consumption from coffee and energy drinks led to a slight decrease in deep sleep duration and an increase in awakenings, the wearable data did not show any significant deterioration in sleep among the group of young, healthy adults. However, caffeine appeared to negatively affect the subjective perception of sleep. The resulting health effects remain unclear, and further research through randomized trials is needed to clarify these findings.

## Figures and Tables

**Figure 1 nutrients-17-01503-f001:**
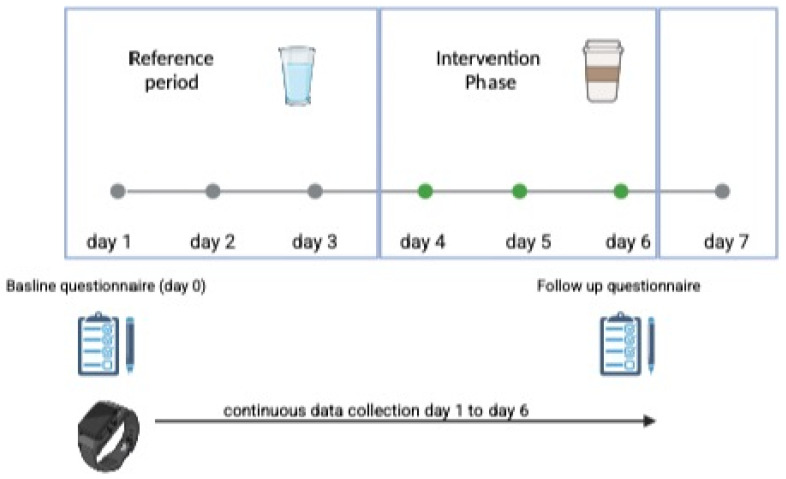
Overview Sleep Smart study (Created in Biorender. Schlichtiger, J. (2025) https://BioRender.com/4y7ccff).

**Figure 2 nutrients-17-01503-f002:**
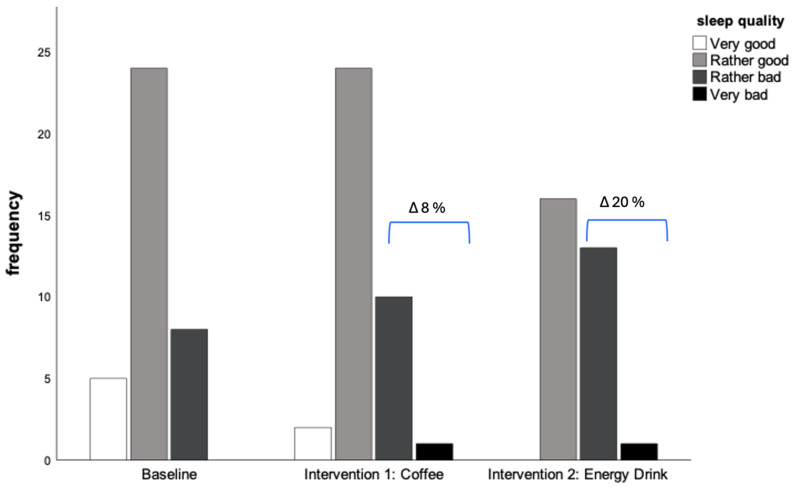
Changes in self-reported sleep quality. ∆: Delta refers to the difference in participant indication of a Rather bad OR Bad Sleep Quality during Intervention 1, 2 vs. Baseline.

**Table 2 nutrients-17-01503-t002:** (**A**): Secondary outcomes (wearable data), (**B**)**:** Secondary outcomes (self-assessment of sleeping measures), (**C**): Secondary outcomes (self-assessment of sleep quality).

(A)							
	Coffee	Ref_C_	Coffee vs. Ref_C_ *p*-Value	Energy Drink	Ref_E_	Energy Drink vs. Ref *p*-Value	Coffee vs. Energy Drink *p*-Value
Wearable data [mean ± sd]							
Light sleep [min]	242.3 ± 4.0	255.7 ± 4.3	0.308	237.5 ± 4.0	261.3 ± 4.4	0.086	0.095
Deep sleep [min]	189.3 ± 3.2	198.7 ± 3.3	0.562	192.3 ± 3.2	195.8 ± 3.3	0.733	0.790
Awake [min]	22.7 ± 0.4	23.4 ± 0.4	0.587	21.3 ± 0.4	18.9 ± 0.3	0.241	0.604
Duration to sleep [min]	2.8 ± 0.1	2.9 ± 0.1	0.803	2.5 ± 0.04	2.4 ± 0.04	0.895	0.062
Duration to wake up [min]	4.2 ± 0.1	4.0 ± 0.1	0.920	4.4 ± 0.1	2.7 ± 0.1	0.161	0.393
Average heart rate [bpm]	62.2 ± 8.5	62.1 ± 10.2	0.650	62.6 ± 9.4	62.3 ± 8.4	0.787	0.934
(**B**)							
	**Baseline**	**Coffee**		**Energy Drink**	**Baseline vs. Coffee vs. Energy Drink** ***p*-Value**		
Time to fall asleep							
mean ± SD [min]	23.5 ± 15.5	27.4 ± 33.2		27.3 ± 25.4	0.559		
(**C**)							
		**Baseline**	**Coffee**	**Energy Drink**	**Baseline vs. Coffee vs. Energy Drink** ***p*-Value**		
Self-assessed sleeping quality							
Very good		13.5 (5)	5.4 (2)	-	0.173		
Rather good		64.9 (24)	64.9 (24)	53.3 (16)			
Rather bad		21.6 (8)	27.0 (10)	43.3 (13)			
Very bad		-	2.7 (1)	3.3 (1)			
PSQI							
Mean ± SD			6.7 ± 2.6	6.5 ± 2.6	0.948		

Ref_C_ reference period (days 1 to 3) during Intervention period coffee drinking; Ref_E_ reference period (days 1 to 3) during Intervention periodenergy drink consumption; SD: standard deviation; metric data: Wilcoxen Test Statistic; categorical data: Chi2 Test; Bonferroni correction for multiple testing was performed; PSQI: Pittsburgh Sleep Quality Index.

**Table 3 nutrients-17-01503-t003:** Comparison of self-assessment after intervention vs. baseline.

	Baseline *n* = 37	Coffee *n* = 37	Energy Drinks *n* = 31	*p*-Value
Problems to fall asleep				
Never	16.2 (6)	18.9 (7)	16.1 (5)	0.317
Rarely	40.5 (15)	37.8 (14)	19.4 (6)	
Sometimes	29.7 (11)	16.2 (6)	32.3 (10)	
Often	10.8 (4)	24.3 (9)	32.3 (10)	
Always	2.7 (1)	2.7 (1)	-	
Wake up at night				
Never	40.5 (15)	27.0 (10)	38.7 (12)	0.546
Rarely	35.1 (13)	32.4 (12)	22.6 (7)	
Sometimes	18.9 (7)	24.3 (9)	19.4 (6)	
Often	5.4 (3)	13.5 (5)	19.4 (6)	
Always	-	2.7 (1)	-	
Problems to feel Rested				
Never	-	8.1 (3)	-	0.014
Rarely	13.5 (5)	10.8 (4)	22.6 (7)	
Sometimes	13.5 (5)	27.0 (10)	41.9 (13)	
Often	59.5 (22)	37.8 (14)	19.4 (6)	
Always	13.5 (5)	16.2 (6)	16.1 (5)	
Sleep medication				
Never	86.5 (32)	89.2 (33)	90.0 (27)	0.676
Less than once aweek	5.4 (2)	5.4 (2)	10.0 (3)	
At least once a week	8.1 (3)	5.4 (4)		
Problems to stay awake during daytime				
Never	51.4 (19)	54.1 (20)	54.8 (17)	0.338
Less than one aweek	35.1 (13)	37.8 (14)	16.7 (5)	
At least once a week	13.5 (5)	8.1 (3)	26.7 (8)	
Problems to manage daily routine				
No Problems	27.0 (10)	32.4 (12)	26.7 (8)	0.965
Rarely	40.5 (15)	40.5 (15)	33.3 (10)	
Sometimes	29.7 (11)	24.3 (9)	36.7 (11)	
Often	2.7 (1)	2.7 (1)	3.3 (1)	

Categorical data: Chi2 Test.

## Data Availability

The raw data supporting the conclusions of this article will be made available by the authors upon request.

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
