# Peer review of "Effects of Caffeine Intake on Self-Administered Sleeping Quality and Wearable Monitoring of Sleep in a Cohort of Young Healthy Adults"

_nutrients, 2025, doi:10.3390/nu17091503_

Round 1
Reviewer 1 Report
Comments and Suggestions for Authors
Overall, an article exploring the effects of coffee vs. energy drink on sleep metrics (objective and subjective). The article should compare these rather than make any statements about effects as no control exists. I commend the authors on a needed area of research.
Revise the abstract to include specific numerical results and statistical outcomes
Remove all oddly placed dashes throughout the abstract and introduction
Reformat the methods section by moving the full list of survey questions and smartwatch-tracked variables into an appendix, right now it just looks like a copy-paste version of a consent form (results look the same)
Clearly explain how caffeine dosage (240 mg) was verified and standardized across coffee and energy drink conditions
Clarify whether caffeine intake earlier in the day was recorded or controlled for across participants, also it would be important to compare self report sleep with wearable data, to determine if participant self report is even remotely accurate
Clarify how participants were assigned to the coffee or energy drink group
Improve the readability and formatting of Tables 1-2, ensuring clear comparison between groups and timepoints (it was off the page)
In all tables, do not asterisk the label "p-value" in the header row; instead, only use asterisks to denote statistically significant p-values in the body of the table
Clarify if and why any wearable-derived metrics were statistically or clinically meaningful
Expand the discussion of the mismatch between objective sleep data and subjective sleep perception, as this is a key finding of the study
Include a brief evaluation of the accuracy and validation status of the wearable used in the study
Explore why energy drinks produced a greater perceived decline in sleep quality despite identical caffeine doses compared to coffee, possibly due to other ingredients or psychosocial factors
Acknowledge that other unmeasured variables such as stress may have influenced sleep
The biggest limitation was the lack of a control group, we can compare 1 vs the other, but how do we know that this is any different than a normal non-dosage night, I understand in theory caffeine should result in impaired sleep but we don't really know without a control and should simply compare conditions rather than stating that either impaired sleep
Author Response
Comment 1: The article should compare these rather than make any statements about effects as no control exists. I commend the authors on a needed area of research.
We thank the reviewer for this important observation and the positive feedback regarding the relevance of our research. We fully agree that, given the absence of a separate control group, our study design allows only for comparisons within the intervention groups and relative to the baseline period, rather than for definitive statements about causal effects. In response to this comment, we have carefully revised the wording throughout the manuscript, particularly in the Abstract, Discussion, and Conclusions sections, to clearly emphasize the comparative and observational nature of our findings. We have reframed our statements accordingly to avoid implying causal effects and to better reflect the exploratory character of our analyses.
Comment 2: Revise the abstract to include specific numerical results and statistical outcomes.
We agree with this comment and provide data within the results section of the abstract [see abstract, p 1].
Comment 3: Remove all oddly placed dashes throughout the abstract and introduction.
We carefully revised the formation of the manuscript; changes were included as suggested.
Comment 4: Reformat the methods section by moving the full list of survey questions and smartwatch-tracked variables into an appendix, right now it just looks like a copy-paste version of a consent form (results look the same)
Changes were included as suggested; the variables sets were included in the appendix, as refered within the text [see methods, p. 4-5 and Appendix].
Comment 5: Clearly explain how caffeine dosage (240 mg) was verified and standardized across coffee and energy drink conditions.
We thank the reviewer for pointing out the need for greater clarity regarding caffeine dosage verification. We agree that the original description of the validation process was not sufficiently detailed. In response, we have revised and expanded the Methods section [see methods, p. 3 – Study Setting] to better explain the standardization and verification of caffeine intake across both intervention groups. Participants were provided with a detailed nutrition diary listing various coffee and energy drink options with their corresponding caffeine contents. They were instructed to document the type, volume, and time of each consumption daily. Study personnel reviewed these diaries to confirm adherence, and no participants exceeded or fell short of the specified daily intake of 240 mg of caffeine. These clarifications have been incorporated into the revised manuscript.We agree with Reviewer 1 that the section on the validation of the daily caffeine dosage using a food diary was not sufficient; we made adjustments to the methods section and included a more detailed explanation.
Comment 6: Clarify whether caffeine intake earlier in the day was recorded or controlled for across participants.
We thank the reviewer for this important point. As noted, participants' habitual caffeine intake earlier in the day (e.g., morning coffee consumption) was maintained throughout the intervention period and was not modified or controlled. This approach was chosen to preserve participants' usual routines and to avoid introducing additional confounders. Habitual caffeine intake was recorded at baseline and treated as a constant variable during the intervention. We agree that this aspect was not sufficiently described in the original manuscript and have now clarified it in the revised Methods section [see methods, p. 3 – Study Setting].
Comment 7: Also it would be important to compare self report sleep with wearable data, to determine if participant self report is even remotely accurate
We thank the reviewer for this very valuable comment. While comparing self-reported sleep duration with wearable-derived data would indeed provide useful insights into the accuracy of subjective assessments, this was not the primary aim of our study. Furthermore, the study was neither specifically designed nor statistically powered to evaluate the agreement or validity between the two measurement methods. Our focus was to assess changes within each method separately, rather than to directly compare them.
Comment 8: Clarify how participants were assigned to the coffee or energy drink group
Study subjects were allowed to self-assign for an intervention group to increase coherence and compliance. Adjustments were made as suggested to clarify [see methods, p. 3 – Study Setting].
Comment 9: Improve the readability and formatting of Tables 1-2, ensuring clear comparison between groups and timepoints (it was off the page)
Adjustments to the tables were made to improve the readability [see results, p. 7 and 8].
Comment 10: In all tables, do not asterisk the label "p-value" in the header row; instead, only use asterisks to denote statistically significant p-values in the body of the table
Changes were included as suggested [see results, p. 7 to 9].
Comment 11: Clarify if and why any wearable-derived metrics were statistically or clinically meaningful
We thank the reviewer for this valuable comment. In the revised Discussion section, we have emphasized more clearly the clinical and scientific potential of wearable-derived sleep data. While no statistically significant differences were observed for most sleep parameters, we discussed the subtle trends in deep sleep reduction and increased awakenings. Additionally, we addressed the relevance and limitations of wearable devices in detecting minor changes and their potential role in future large-scale, real-world investigations.
Comment 12: Expand the discussion of the mismatch between objective sleep data and subjective sleep perception, as this is a key finding of the study
We thank the reviewer for this important comment. We agree that the observed mismatch between subjective sleep perception and objective wearable-derived sleep parameters represents a key finding of the SleepSmart study. We have expanded the corresponding paragraph in the Discussion section to highlight the clinical and research relevance of this discrepancy, and to discuss possible underlying mechanisms such as perception bias, unmeasured psychosocial factors, or additional influences beyond physiological sleep structure [see discussion, p. 12].
Comment 13: Include a brief evaluation of the accuracy and validation status of the wearable used in the study.
We thank the reviewer for this helpful comment. The Withings ScanWatch has been validated for heart rate monitoring at rest and for basic sleep tracking parameters. We agree that this important information was not sufficiently emphasized in the original version of the manuscript. We have therefore added a corresponding description of the validation status and limitations of the device to the Methods section [see methods, p. 4].
Comment 14: Explore why energy drinks produced a greater perceived decline in sleep quality despite identical caffeine doses compared to coffee, possibly due to other ingredients or psychosocial factors
We thank the reviewer for this important observation. We agree that other ingredients contained in energy drinks—such as sugar, taurine, and other supplements—could have contributed to the greater perceived decline in sleep quality, despite identical caffeine doses. We have added a corresponding paragraph to the Discussion section addressing these potential effects as well as the possible influence of psychosocial factors. [see discussion, p. 11].
Comment 15: Acknowledge that other unmeasured variables such as stress may have influenced sleep
We thank the reviewer for this important comment. We acknowledge that unmeasured variables, such as stress, diet, and mental well-being, could have influenced sleep outcomes and contributed to the observed mismatch between subjective and objective sleep measures. We have adjusted the relevant paragraph in the Discussion to emphasize the potential impact of these unassessed factors. [see discussion, p. 12].
Comment 16: The biggest limitation was the lack of a control group, we can compare 1 vs the other, but how do we know that this is any different than a normal non-dosage night, I understand in theory caffeine should result in impaired sleep but we don't really know without a control and should simply compare conditions rather than stating that either impaired sleep
We agree with the reviewer that the lack of a randomized control group represents a major limitation of our study. We have revised the Discussion section to clearly emphasize the exploratory nature of our findings and to acknowledge that, without a control group, we can only compare the two intervention conditions rather than draw definitive conclusions regarding impairment relative to normal sleep. Additionally, we have outlined the necessity for future randomized studies with appropriate control groups to validate and expand on our observations. [see discussion, p. 12]
Reviewer 2 Report
Comments and Suggestions for Authors
A very interesting study that used both smartwatch data and questionnaire responses to characterize sleep. The goal of the SleepSmart study was to assess the impact of caffeine consumption from both coffee and energy drinks on sleep patterns. If the goal is to examine the impact of caffeine consumption, then I would expect, in addition to information on the average amount of caffeine consumed in both coffee and Energy Drink, to examine the relationship between the amount of caffeine consumed and the length and quality of sleep. This would allow us to assess whether comparing the importance of coffee or Energy Drink for sleep is really a comparison of two drinks or two doses of caffeine. I would also like to point out that the days when the subjects did not consume coffee or Energy Drink could have been quite unpleasant for the subjects who were regular consumers of these drinks. The entire description of the study can be reversed and one can wonder how periodic cessation of caffeine consumption affects sleep parameters (I recommend reading the article "O'Callaghan F, Muurlink O, Reid N. Effects of caffeine on sleep quality and daytime functioning. Risk Manag Healthc Policy. 2018 Dec 7;11:263-271. doi: 10.2147/RMHP.S156404")
Author Response
Comment 1: If the goal is to examine the impact of caffeine consumption, then I would expect, in addition to information on the average amount of caffeine consumed in both coffee and Energy Drink, to examine the relationship between the amount of caffeine consumed and the length and quality of sleep. This would allow us to assess whether comparing the importance of coffee or Energy Drink for sleep is really a comparison of two drinks or two doses of caffeine. I would also like to point out that the days when the subjects did not consume coffee or Energy Drink could have been quite unpleasant for the subjects who were regular consumers of these drinks.
We thank the reviewer for this thoughtful comment. We agree that the baseline caffeine intake prior to the intervention was not sufficiently described and have made adjustments in the Methods section to clarify this. [see methods, p. 3 – Study Setting]
Specifically, the participants' usual caffeine intake (e.g., coffee consumption earlier in the day) was maintained throughout the study to serve as a constant variable. This approach aimed to avoid introducing confounding effects due to changes in habitual consumption patterns during the intervention phase. Thus, the comparison focused solely on the additional, standardized evening caffeine intake via coffee or energy drinks.
We also acknowledge the reviewer’s point regarding potential withdrawal effects on days without coffee or energy drink consumption. However, since participants were allowed to maintain their normal daytime caffeine routines during the baseline and intervention periods, significant withdrawal symptoms during the study are unlikely.
Comment 2: The entire description of the study can be reversed and one can wonder how periodic cessation of caffeine consumption affects sleep parameters (I recommend reading the article "O'Callaghan F, Muurlink O, Reid N. Effects of caffeine on sleep quality and daytime functioning. Risk Manag Healthc Policy. 2018 Dec 7;11:263-271. doi: 10.2147/RMHP.S156404")
We sincerely thank the reviewer for this valuable comment and for suggesting the article by O'Callaghan et al., which we have carefully reviewed. We fully agree that the question of how periodic cessation of caffeine consumption affects sleep parameters is highly relevant.
However, based on the design and available data of the SleepSmart study, conclusions regarding the effects of caffeine withdrawal cannot be drawn. Our focus was on comparing sleep parameters during habitual intake versus standardized additional evening intake, without implementing or analyzing a structured cessation phase.
In the revised Discussion section, we have emphasized this limitation more clearly and outlined the need for future studies specifically investigating the effects of caffeine withdrawal and periodic cessation as an extension of our current work.
Round 2
Reviewer 1 Report
Comments and Suggestions for Authors
The Authors have addressed the majority of concerns except formatting. Again, there are still oddly placed hyphens (nobody writes like that, so why are they there?) in the middle of text, odd indenting, and weirdly placed lists such as "criteria." Please restructure these scientifically in paragraph form. Simply put these things into your own words/structure and update.
Reviewer 2 Report
Comments and Suggestions for Authors
In my opinion, the authors, following the reviewers' suggestions, have made significant improvements to the text. I have no further comments and suggest publishing the text without changes.